

# Barefoot running does not affect simple reaction time: an exploratory study

Nicholas J. Snow[1], Jason F.L. Blair[2], Graham Z. MacDonald[3], Jeannette M. Byrne[2] and Fabien A. Basset[2]

[1] Faculty of Medicine, Memorial University of Newfoundland, St. John's, Newfoundland and Labrador, Canada
[2] School of Human Kinetics and Recreation, Memorial University of Newfoundland, St. John's, Newfoundland and Labrador, Canada
[3] Human Performance Laboratory, Faculty of Kinesiology, University of Calgary, Calgary, Alberta, Canada

Corresponding authors
Jeannette M. Byrne, jmbyrne@mun.ca
Fabien A. Basset, fbasset@mun.ca

## ABSTRACT

**Background.** Converging evidence comparing barefoot (BF) and shod (SH) running highlights differences in foot-strike patterns and somatosensory feedback, among others. Anecdotal evidence from SH runners attempting BF running suggests a greater attentional demand may be experienced during BF running. However, little work to date has examined whether there is an attentional cost of BF versus SH running.

**Objective.** This exploratory study aimed to examine whether an acute bout of BF running would impact simple reaction time (SRT) compared to SH running, in a sample of runners naïve to BF running.

**Methods.** Eight male distance runners completed SRT testing during 10 min of BF or SH treadmill running at 70% maximal aerobic speed ($17.9 \pm 1.4\,\mathrm{km\,h^{-1}}$). To test SRT, participants were required to press a hand-held button in response to the flash of a light bulb placed in the center of their visual field. SRT was tested at 1-minute intervals during running. BF and SH conditions were completed in a pseudo-randomized and counterbalanced crossover fashion. SRT was defined as the time elapsed between the light bulb flash and the button press. SRT errors were also recorded and were defined as the number of trials in which a button press was not recorded in response to the light bulb flash.

**Results.** Overall, SRT later in the exercise bouts showed a statistically significant increase compared to earlier ($p < 0.05$). Statistically significant increases in SRT were present at 7 min versus 5 min ($0.29 \pm 0.02$ s vs. $0.27 \pm 0.02$ s, $p < 0.05$) and at 9 min versus 2 min ($0.29 \pm 0.03$ s vs. $0.27 \pm 0.03$ s, $p < 0.05$). However, BF running did not influence this increase in SRT ($p > 0.05$) or the number of SRT errors ($17.6 \pm 6.6$ trials vs. $17.0 \pm 13.0$ trials, $p > 0.05$).

**Discussion.** In a sample of distance runners naïve to BF running, there was no statistically significant difference in SRT or SRT errors during acute bouts of BF and SH running. We interpret these results to mean that BF running does not have a greater attentional cost compared to SH running during a SRT task throughout treadmill running. Literature suggests that stride-to-stride gait modulation during running may occur predominately via mechanisms that preclude conscious perception, thus potentially attenuating effects of increased somatosensory feedback experienced during BF running. Future research should explore the present experimental paradigm in a larger sample using over-ground running trials, as well as employing different tests of attention.

## BACKGROUND

Despite a considerable amount of research focusing on footwear's role in injury prevention (*Gallant & Pierrynowski, 2014*), injury rates have remained constant over the past 40 years (*Lieberman, 2012*). Approximately 85% of runners experience running-related musculoskeletal injuries throughout their running career, and 30–70% of runners are treated for these injuries annually (*Nielsen et al., 2012*). This high prevalence of running-related injuries has led to investigations into the mechanisms contributing to their etiology (*Hreljac, 2005*), and to alternative solutions beyond the classic recommendation of a change in footwear characteristics. In this context, barefoot (BF) running has been proposed as an alternative solution (*Murphy, Curry & Matzkin, 2013*), and has gained substantial traction in the public (*Hryvniak, Dicharry & Wilder, 2014*). Indeed, recent literature describes several distinct differences between shod (SH) and BF running. Of interest are changes in foot-strike patterns (*Divert et al., 2005*; *Lieberman et al., 2010*; *Squadrone & Gallozzi, 2009*), movement kinematics (*Squadrone & Gallozzi, 2009*), and muscle activation (*Snow, Basset & Byrne, 2016*; *Von Tscharner, Goepfert & Nigg, 2003*).

BF running has been promoted as a method to increase foot plantar sensation (*Robbins, Gouw & Hanna, 1989*; *Robbins et al., 1993*; *Robbins & Hanna, 1987*), feedback that is believed to be masked during SH running (*Robbins et al., 1993*; *Robbins, Waked & McClaran, 1995*). Improved plantar sensory feedback associated with BF running has been suggested to modify kinematic variables (e.g., stride length, stride frequency) in a way that ultimately alters foot-strike patterns (*Daoud et al., 2012*; *Lieberman, 2012*) to reduce plantar pain and the risk of injury due to repetitive impact (e.g., during SH running with a heel-to-toe foot-strike; *Lieberman et al., 2010*). Given running involves chronic repetitive movement, it is no surprise that kinematic differences in the running gait can greatly impact one's risk of injury (*Daoud et al., 2012*; *Hreljac, 2005*). Furthermore, running is associated with a high risk of acute injury due to trips, falls, and sprains (*Hsu, 2012*; *Knobloch, Yoon & Vogt, 2008*); thus, the prevalence of both acute and chronic injuries in running demonstrate that efficient cognitive processing and rapid reaction to perturbations or obstacles is imperative during running.

Prior work has suggested that the speed of basic information processing is a valid indicator of higher cognitive function (*Kail & Salthouse, 1994*; *Woods et al., 2015*), and that decrements in higher-order cognitive operations can be reflected by diminished performance in simple tasks evaluating the speed of processing (*Kail & Salthouse, 1994*; *Woods et al., 2015*). Simple reaction time (SRT) is a task frequently used to measure speed of processing (*Woods et al., 2015*). Tests of SRT often involve making a physical response (e.g., pressing a button) to the presentation of a visual stimulus (e.g., light bulb flash). SRT is thus defined as minimum amount of time needed to respond to the stimulus (*Woods et al., 2015*). Furthermore, when an SRT task is simultaneously combined with another demanding situation (e.g., exercise), it causes a dual-task situation (*Watanabe & Funahashi, 2017*).

In the dual-task, poorer performance often results in one or both tasks, relative to when they are preformed alone (*Watanabe & Funahashi, 2017*). This dual-task interference effect has been established as an important indicator of humans' limited capacity for information processing (*Watanabe & Funahashi, 2017*). Past evidence has demonstrated that concurrent acute exercise can result in decrements in performance on SRT tasks (*Brisswalter et al., 1997*; *McMorris & Keen, 1994*). For example, a dual-task interference effect was observed during rhythmic exercise when a non-preferred foot cadence was adopted (*Brisswalter et al., 1995*; *Collardeau, Brisswalter & Audiffren, 2001*). This evidence suggests that maintaining a novel stride frequency requires considerable attention, sufficient to cause participants' SRT performance to deteriorate. If a non-preferred stride frequency can alter SRT, then SRT could also be compromised during BF running, which promotes greater stride frequency (*Divert et al., 2005*) and sensory feedback that causes participants to pay more attention to their foot-strike (*Abernethy, Hanna & Plooy, 2002*; *Brisswalter et al., 1995*; *Hanson, Whitaker & Heron, 2009*). In line with this idea, a recent study suggested that BF running requires a greater level of attention than SH, due to a greater need to focus on placing footfalls on the ground (*Alloway et al., 2016*).

The literature also indicates that individuals with worse reaction time performance are prone to increased ankle instability (*Konradsen & Ravn, 1990*; *Konradsen & Ravn, 1991*), and may be at a greater risk of falls (*Richardson et al., 2017*) and acute lower-limb injuries such as ankle sprains (*Beynnon et al., 2001*; *Murphy & Connolly, 2003*; *Willems et al., 2005*). Thus, given the greater sensory feedback associated with BF running (*Robbins, Gouw & Hanna, 1989*; *Robbins & Hanna, 1987*), it is possible that BF running requires a greater attentional demand relative to SH running, potentially leading to decrements in SRT and increased acute injury risk due to sprains, collisions, stumbles, or falls.

Therefore, the purpose of the present study was to determine if an acute bout of BF running influenced SRT compared to SH running, in a sample of competitive distance runners naïve to BF running. We hypothesized that SRT would be increased during BF running, relative to SH running at a similar exercise intensity.

## MATERIALS & METHODS

This study was approved by the Interdisciplinary Committee on Ethics in Human Research at the Memorial University of Newfoundland (ethics approval number: 20130246-HK), with informed consent being gathered in accordance with the principles outlined by the Declaration of Helsinki.

Research participants were recruited using posters distributed throughout the Memorial University of Newfoundland campus athletic facilities and Physical Education Building, as well as local running retail outlets in the community of St. John's, NL, Canada. A standard recruitment email was also distributed to local running clubs and the Memorial University of Newfoundland's varsity running team.

### Participants

Inclusion criteria consisted of the following: (i) male participants; (ii) aged between 19 and 30 years; (iii) experienced runners, operationally defined as a function of running

experience ($\geq$2 years of active running training), frequency ($\geq$4 days of running per week), and duration ($\geq$30 min per day spent running); (iv) competitive racing experience ($\leq$18 min to complete 5 km; $\leq$40 min to complete 10 km); (v) experience training at a high intensity ($\leq$4 min km$^{-1}$ running pace, $\geq$1 day per week); (vi) free of any chronic illnesses including cardiometabolic, neurological, or psychiatric diagnoses, or neuromuscular and musculoskeletal injuries, for at least 3 months; (vii) active use of a minimalist running shoe ($\leq$4 mm heel-toe drop, measured as the difference between the sole height of the heel and toe of the shoe), with greater than 3 months of experience and at least 3 days of use per week, for at least 30 min per session; and (viii) naïve to BF running.

Twenty-three participants responded to recruitment materials. Of these participants, nine were eligible to participate. One participant withdrew from the study after initial enrollment, due to personal time constraints. Thus, eight adult male distance runners were enrolled in the present study. No *a priori* power calculations were performed to arrive at this sample size; rather, this convenience-sample of eight participants was examined as an exploratory study, to generate data intended to inform further research on the present topic. To limit variability in SRT results, we recruited male participants only, given well-documented sex differences in SRT performance throughout the literature (*Jain et al., 2015*; *Karia et al., 2012*; *Silverman, 2006*; *Woods et al., 2015*). Participants were all competitive runners in the general preparatory phase of their training. Participants were experienced with treadmill running. Participants completed both a short- and long-form Physical Activity Readiness Questionnaire (PAR-Q) (*Canadian Society for Exercise Physiology, 2002*) to ensure that they were injury-free for at least 3 months prior to enrolling in the study, and to screen for any injuries or health conditions that would preclude their inclusion in the study. Participants were instructed to refrain from strenuous exercise at least 36 h prior to testing (*Dannecker & Koltyn, 2014*); to avoid caffeine, alcohol, drugs, or supplements at least 24 h prior to testing; and were required to obtain at least 6 h of sleep the night prior to each testing session.

## Experimental conditions

Each participant was subjected to both BF and SH experimental conditions in a crossover fashion (Fig. 1). For the BF running condition, rubber-gripped toe-socks (Gaiam No Slip Yoga Socks; Gaiam Inc., Boulder, CO, USA) were used. All participants were naïve to BF running and running in the socks provided. As participants had likely not developed the appropriate responses to minimize discomforts associated with true barefoot running (*Lieberman, 2012*; *Robbins et al., 1993*), the toe-socks were used to help minimize potential abrasions or friction burns associated with true barefoot running on a treadmill (*Snow, Basset & Byrne, 2016*). Previous studies have employed similar garments during assessments of BF running (*Divert et al., 2008*; *Snow, Basset & Byrne, 2016*). For the SH running condition, participants were required to bring in a pair of their own running shoes (*Snow, Basset & Byrne, 2016*), to mitigate any negative influences footwear discomfort may have on SRT (*Mündermann et al., 2003*). In accordance with previous studies, these running shoes were >225 g in mass, had a >5 mm heel-toe drop, and were with or without
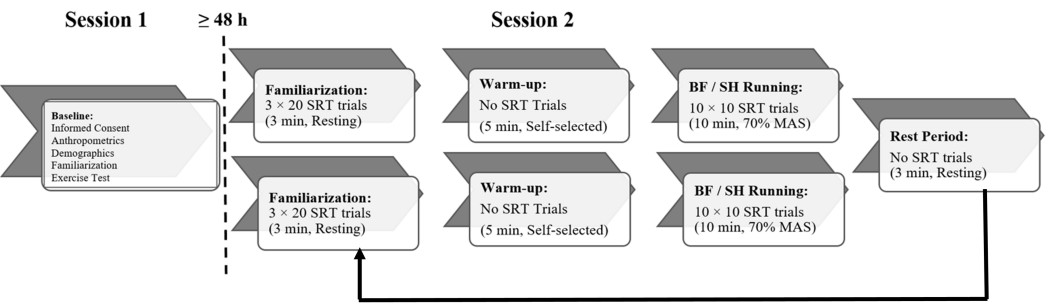

**Figure 1** **Schematic of experimental protocol for simple reaction time (SRT) testing.** Briefly, eight participants attended the laboratory on Day 1 for informed consent, measurement of anthropometrics (i.e., body mass, height), collection of demographic information, experimental setup familiarization, and incremental running exercise test for determination of maximal $O_2$ uptake ($\dot{V}O_{2max}$) and maximal aerobic speed (MAS). On Day 2, participants completed SRT testing under both barefoot (BF) and shod (SH) running conditions. The order of conditions was randomized and counterbalanced across the sample. After completion of one condition, participants crossed-over into the other (black arrow). Prior to commencing either the BF or SH running SRT trials, participants completed three, 20-stimulus SRT familiarization trials while standing at rest. Both SH and BF running conditions were preceded by a 5-min warm-up at participants' self-selected treadmill speed. Participants completed both BF and SH running conditions for 10 min each at 70% MAS. During these running trials, 10 SRT stimuli were presented every minute. A 3-min seated rest period separated conditions; SRT was not tested during warm-up or rest periods.

**Table 1** **Participant characteristics (Mean ± SD).**

| Age | Body Mass | Height | BMI | $HR_{max}$ | % Age-predicted $HR_{max}$ | $\dot{V}O_{2max}$ | MAS |
|---|---|---|---|---|---|---|---|
| (yr) | (kg) | (cm) | (kg m$^{-2}$) | (bpm) | (%) | (mL min$^{-1}$ kg$^{-1}$) | (km h$^{-1}$) |
| 25.1 ± 3.7 | 78.4 ± 8.9 | 180.7 ± 7.8 | 24.0 ± 2.0 | 191 ± 4 | 96.5 ± 6.5 | 61.4 ± 6.7 | 17.9 ± 1.4 |

Notes.

BMI, body mass index; $HR_{max}$, maximal heart rate; $\dot{V}O_{2max}$, maximal oxygen uptake; MAS, maximal aerobic speed.

medial arch support or impact attenuation features (*Esculier et al., 2015*; *Rixe, Gallo & Silvis, 2012*). Participants did not wear minimalist footwear during the experiment.

## Experimental set-up and protocol

The experimental protocol was administered over two testing sessions separated by ≥48 h, as depicted in Fig. 1. All testing sessions were conducted in the morning.

### Testing session one

During this baseline session, participants' informed consent, anthropometrics (i.e., body mass, height), and demographic information were collected first (Tables 1 and 2). Participants were then familiarized with the experimental conditions and set-up, before completing an incremental treadmill exercise test. All running trials were conducted on a Cybex 750T motorized treadmill (Cybex International, Inc., Medway, MA, USA) set at a constant 1% grade to account for air-resistance experienced when running outdoors (*Jones & Doust, 1996*).
**Table 2  Participant training profile (Mean ± SD).**

| Training experience (yr) | Training sessions (n wk$^{-1}$) | Interval training (>75% $\dot{V}O_{2max}$) (n wk$^{-1}$) | Training load (km wk$^{-1}$) | Minimalist shoe experience (mo) | Minimalist training load (km wk$^{-1}$) | 10 km personal best (min:s) |
|---|---|---|---|---|---|---|
| 3.1 ± 2.1 | 8.1 ± 3.5 | 1.9 ± 0.8 | 90.0 ± 44.7 | 11.5 ± 11.4 | 72.3 ± 62.0 | 37:26 ± 2:50 |

Notes.

$\dot{V}O_{2max}$, maximal oxygen uptake.

For familiarization, participants were instructed to run at a self-selected treadmill speed in both their standard running footwear (SH running, $11.0 \pm 0.5$ km h$^{-1}$) and the toe-socks provided (BF running, $10.7 \pm 0.8$ km h$^{-1}$), for 2.5 min each, in random order. Heart rate (HR) was not assessed during these running trials. Furthermore, these running trials were not intended to habituate participants to the footwear conditions in advance of SRT testing, but to provide them with an expectation of how each running condition felt, to aid in their decision to either continue in the subsequent study session or withdraw. Immediately following this familiarization session, exercise testing was conducted, and consisted of an incremental treadmill test to exhaustion. The treadmill test was used to determine maximal O$_2$ uptake ($\dot{V}O_{2max}$) and maximal aerobic speed (MAS), defined as the participants' running speed at $\dot{V}O_{2max}$ (*Basset, Chouinard & Boulay, 2003*). Participants were instructed to wear their preferred footwear for exercise testing, given prior evidence that running economy is greater when participants wear shoes with a higher comfort rating (*Luo et al., 2009*). The incremental test started at a treadmill speed of 7.0 km h$^{-1}$ and was increased by 1.0 km h$^{-1}$ every 2 min until participants reached volitional exhaustion (*Leger & Boucher, 1980*). To ensure participants reached $\dot{V}O_{2max}$ upon volitional exhaustion (as opposed to peak $\dot{V}O_2$), they recovered for 5 min at walking speed prior to the treadmill speed being increased to 105% of MAS (*Rossiter, Kowalchuk & Whipp, 2006*). Participants were instructed to maintain 105% of MAS until they reached their limit of tolerance (*Rossiter, Kowalchuk & Whipp, 2006*). All participants reached $\dot{V}O_{2max}$ during the treadmill test.

Exercise metabolic rate of the incremental test was recorded with an indirect calorimetry system (AEI Technologies, Inc., Pittsburgh, PA, USA). Oxygen uptake ($\dot{V}O_2$), carbon dioxide ($\dot{V}CO_2$), breathing frequency (B$f$), and tidal volume (V$_T$) were continuously collected with an automated open-circuit gas analysis system using O$_2$ and CO$_2$ analyzers (Model S-3A and Anarad AR-400; Ametek, Pittsburgh, PA), and a pneumo-tachometer (Model S-430; Vacumetrics/Vacumed Ltd., Ventura, CA) with a 4.2 L mixing chamber. Respiratory exchange ratio (RER) and minute ventilation ($\dot{V}_E$) were calculated as the quotient of $\dot{V}CO_2$ on $\dot{V}O_2$ and as the product of B$f$ by V$_T$, respectively. Online HR data were wirelessly transmitted to the AEI indirect calorimetric system with a telemetric Polar HR monitor (Polar Electro, Oy, Finland). Prior to testing, volume and gas analyzers were calibrated with a 3.0 L calibration syringe and medically certified O$_2$ and CO$_2$ calibration gases that were 16% O$_2$ and 4% CO$_2$, respectfully. The data were online digitalized from an A/D card to a computer for monitoring the metabolic rate (AEI Metabolic System Software; AEI Technologies, Inc., Pittsburgh, PA, USA). Results of the $\dot{V}O_{2max}$ testing were used to determine running speeds implemented during later experimental trials.

### Testing session two

On day two of testing participants completed two, 10-min running trials, one BF and the other SH. During each of the 10-min trials participants' SRT was tested 10 times every minute. Participants' HR was not measured during these running trials. Both BF and SH running conditions were completed at 70% of the participants' MAS, which was believed to coincide with a level of physiological arousal that optimizes SRT (*Brisswalter, Collardeau & René, 2002*; *Collardeau, Brisswalter & Audiffren, 2001*). Past work suggests that exercise at such an intensity and duration can facilitate cognitive processes during exercise (*Lambourne & Tomporowski, 2010*; *Tomporowski, 2003*). This exercise intensity was selected to emphasize the potential effect that footwear condition would have on SRT, without the contaminating effect of fatigue or inappropriate exercise intensity. Conditions were pseudo-randomized and counterbalanced across the study sample, such that the order of BF and SH running was reversed for every other participant, to prevent an order effect of running conditions on SRT performance.

SRT was defined as the time required to press a hand-held button in response to the flash of a 40 W soft-white light bulb placed in the center of the participants' visual field. The SRT device used presently was developed at the Memorial University of Newfoundland and has been used in previous research examining exercise effects on SRT (*Behm et al., 2004*). The between- and within-session reliability of SRT measurements was shown by intra-class correlation coefficients of 0.60 and 0.79 (moderate to good reliability), respectively, with no statistically significant ($p > 0.05$) differences between test and re-test values (*Behm et al., 2004*). Participants held the button apparatus in their dominant hand during all SRT procedures. The apparatus was also affixed to participants' dominant wrist with a fabric-lined Velcro strap, to prevent dropping. To eliminate any auditory distractions, participants wore ear-plugs along with a noise-cancelling headset. To eliminate any visual distractions, barricades restricted participant's peripheral field of view. Both triggering of the light bulb (SRT stimulus) and the button-press (SRT response) were recorded at 2,000 Hz, sampled using a BIOPAC MP100 biological amplifier, and displayed using AcqKnowledge 3.9.1 software (BIOPAC Systems, Inc., Goleta, CA, USA). All SRT data were stored offline on a computer and later pre-processed using AcqKnowledge and Microsoft Excel software.

Prior to commencing each running condition, participants completed three, 20-stimulus, SRT familiarization trials while standing on the treadmill at rest (*Brisswalter et al., 1997*; *Brisswalter et al., 1995*). Prior to both SRT familiarization and exercise periods, participants were instructed to focus on a small target just below the light bulb and to respond quickly and vigilantly to the presented stimuli. Following SRT familiarization, participants warmed up for 5 min at their self-selected treadmill speed (SH running, $11.0 \pm 0.5$ km h$^{-1}$; BF running, $10.7 \pm 0.8$ km h$^{-1}$), completed the required condition (i.e., BF or SH), and then rested for 3 min prior to repeating the protocol, completing the second condition. In combination with the subsequent familiarization SRT trials, participants were inactive (i.e., not running) for a total of 6 min. This time allowed for recovery of HR between running conditions (*Saltin et al., 1968*), with the intention of not influencing cognitive performance under the subsequent condition (*Lambourne & Tomporowski, 2010*). During each 10-min

condition (i.e., BF and SH), SRT testing was administered in blocks of 10 SRT stimuli, delivered over the last 50 s of each minute, with each of the stimuli separated by a random interval to prevent anticipation of subsequent trials (*Schmidt & Lee, 2005*). The above setup was intended to produce a dual-task effect, to examine which running condition would have a greater influence on participants' SRT. To complement the objective SRT information provided we also asked participants to briefly comment on their experience of BF versus SH running.

Video footage was gathered on the right lower-limb, for both SH and BF running conditions to assess stride frequency. A Sony HDR-CX430VB 30 Hz video camera (Sony Computer Entertainment America, San Mateo, CA, USA) was positioned perpendicular to the treadmill at a distance of 1.5 metres and a height of 0.75 metres. Video footage was collected at a 30 Hz frame rate, in accordance with previous literature (*Macpherson et al., 2016*; *Nikodelis et al., 2011*), with a total of 30 running strides per participant being collected for each running condition (i.e., BF and SH). Raw video data were converted to MPEG-4 using Sony PMB software (Sony Computer Entertainment America, San Mateo, CA, USA) for further analysis.

## Data analyses

### $\dot{V}O_{2max}$

Participants' $\dot{V}O_{2max}$ was considered the peak value in $O_2$ uptake using a 30-s moving window average technique. MAS was the corresponding treadmill speed at $\dot{V}O_{2max}$ (*American College of Sports Medicine, 2013*; *Basset, Chouinard & Boulay, 2003*). $HR_{max}$ was defined as the peak HR value obtained during the $\dot{V}O_{2max}$ test (*American College of Sports Medicine, 2013*).

### Stride frequency

The Kinovea (Version 2.0) high-resolution video analysis software platform (http://www.kinovea.org/) was used to determine the frame of foot-contact and toe-off during each 10-s window, for each minute, throughout each 10-min trial, for both the BF and SH running conditions (*Damsted, Nielsen & Larsen, 2015*; *Padulo et al., 2015*). To minimize error in video interpretation, blind cross-checks were performed by two researchers (NJS, JMB). Foot-contact and toe-off were used to determine stride frequency (strides $s^{-1}$) by counting the number of complete strides per 10 s of video data. This number was then multiplied by 6 to provide the final stride frequency estimate (strides $min^{-1}$).

### SRT

Participants' SRT was considered the time difference (in seconds) between the initiation of the SRT stimulus (light bulb) and the completion of the SRT response (button press) (*Magill, 2011*). As such, SRT encompassed participants' overall response time, which is comprised of both reaction time (i.e., time between stimulus presentation and initiation of response) and movement time (i.e., time between response initiation and response completion) (*Magill, 2011*). Therefore, our SRT measure contained a global measure of both stimulus detection (reaction time) and response execution (movement time) but did not distinguish between them. Any SRT trial <0.160 s was considered an anticipated response

and was to be omitted from the data set (*Brisswalter et al., 1997*; *Brisswalter et al., 1995*; *Collardeau, Brisswalter & Audiffren, 2001*). However, zero SRT responses met this criterion, and thus no SRT trials were omitted from the data set on this basis. We also analyzed SRT errors, which consisted of instances where participants did not respond to the stimulus (light bulb), or when SRT trials >1.0 s were recorded (*Woods et al., 2015*). No SRT trials exceeded 1.0 s, and so SRT errors were considered only for those trials that were un-recorded. The number of un-recorded SRT responses were counted by the AcqKnowledge software during each minute and averaged across each total running trial (BF, SH). Remaining SRT trials were averaged over each minute for their respective running conditions (i.e., BF and SH running). Increasing SRT (in seconds) was indicative of an increase in perceptual latency and attentional load.

## Statistical analyses

All data were examined for normality using the Shapiro–Wilk test and visual examination of histogram plots. Because of sensitivity to sample size variations in statistics-based normality tests, as well as the robustness of within-subjects designs to normality violations, we used a stringent significance level of $p < 0.001$ in objective examinations of the data distributions (i.e., Shapiro–Wilk test) (*Gamst, Meyers & Guarino, 2008*; *Mang et al., 2016*).

The effect of BF versus SH running on SRT was tested using a two-way (2 levels $\times$ 10 levels) repeated-measures analysis of variance (rmANOVA) with the factors Condition (BF running, SH running) and Time (1–10 min). To determine if SRT during BF running, SH running, or running in general was different from rest, the average SRT from the familiarization trials, the BF and SH running conditions, and both exercise conditions combined was compared using a one-way (4 levels) rmANOVA with the factor Condition (Familiarization, BF running, SH running, Combined running). *Post hoc* pairwise comparisons were conducted when necessary using the Bonferroni correction. Mean SRT errors (number of trials) were compared across BF and SH running conditions using a paired samples $t$-test. Finally, average stride frequency (strides min$^{-1}$) was compared across conditions (BF running, SH running) using a paired samples $t$-test.

All results are presented as means $\pm$ one standard deviation (SD). Statistical significance was set at $p < 0.05$. Effect sizes (Cohen's $d$) and 95% confidence intervals (95% CI) of effect sizes were calculated for our primary outcome (SRT in seconds) using Microsoft Excel, and interpreted as "trivial" <0.20; "small" 0.20–0.49; "medium" 0.50–0.79; "large" >0.80 (*Cohen, 1988*). All statistical analyses were conducted in SPSS (Version 20; IBM Corporation, Armonk, NY, USA).

## RESULTS

### Data inspection

All data were deemed normally distributed based on Shapiro–Wilk statistics (SRT $W_{(8)} = 0.774 - 0.990$, $p = 0.015 - 0.996$; stride frequency $W_{(8)} = 0.876 - 0.912$, $p = 0.171 - 0.368$) and histogram plot inspection. For SRT data, Mauchly's Test of Sphericity was not statistically significant ($p > 0.05$); thus, sphericity was assumed for interpreting the results of the rmANOVA on SRT values.

## Participants
### Baseline characteristics

Participants were on average $25.1 \pm 3.7$ years of age, with a body mass, height, and body mass index (BMI) of $78.4 \pm 8.9$ kg, $180.7 \pm 7.8$ cm, and $24.0 \pm 2.0$ kg m$^{-2}$, respectively (Table 1). Their training experience ranged from 1 to 8 years ($3.1 \pm 2.1$ yr), and all participants were experienced in using minimalist footwear ($11.5 \pm 11.4$ mo). Participants completed an average of $8.1 \pm 3.5$ training sessions per week, including $1.9 \pm 0.8$ sessions of interval training at $>75\%$ $\dot{V}O_{2max}$. Total training volume was $90.0 \pm 44.7$ km per week, and $65.3 \pm 44.9\%$ of training volume was completed using minimalist footwear ($72.3 \pm 62.0$ km wk$^{-1}$). Average 10 km personal best race time (mm:ss) was $37:26 \pm 2:50$. This information is detailed in Table 2.

### $\dot{V}O_{2max}$ testing

Baseline exercise testing results can be found in Table 1. Participants' $\dot{V}O_{2max}$ was on average $61.4 \pm 6.7$ mL min$^{-1}$ kg$^{-1}$, corresponding to "excellent" fitness (*American College of Sports Medicine, 2013*). In fact, all participants achieved $\dot{V}O_{2max}$ scores in the 95th–99th percentile based on age and sex norms (*American College of Sports Medicine, 2013*). Average $HR_{max}$ was $191 \pm 4$ bpm ($96.5 \pm 6.5\%$ age-predicted $HR_{max}$). Mean maximal aerobic speed (MAS) was $17.9 \pm 1.3$ km h$^{-1}$, while 70% MAS (for BF and SH running SRT trials) was $12.5 \pm 0.9$ km h$^{-1}$.

## SRT

SRT results from BF running, SH running, and combined across conditions are shown in Fig. 2. A statistically significant main effect of Time ($F_{(9,63)} = 3.097$, $p = 0.004$) was present when assessing SRT. Statistically significant increases in SRT were present at 7 min relative to 5 min ($0.29 \pm 0.02$ s vs. $0.27 \pm 0.02$ s, $p < 0.05$, $d = -0.99$, 95% CI [$-1.98$–$+0.09$], large effect), and at 9 min relative to 2 min ($0.29 \pm 0.03$ s vs. $0.27 \pm 0.03$ s, $p < 0.05$, $d = -0.67$, 95% CI [$-1.63$–$+0.37$], moderate effect). There was neither a statistically significant main effect of Condition ($F_{(1,7)} = 1.002$, $p = 0.350$) nor a statistically significant Condition $\times$ Time interaction effect ($F_{(9,63)} = 1.233$, $p = 0.292$). Examination of effect sizes between conditions indicated that overall, BF running had a small negative effect on SRT ($d = -0.32$, 95% CI [$-3.29$–$+2.65$]). The largest negative effect BF running had on SRT was at 8 min ($d = -0.80$, 95% CI [$-1.65$–$+0.06$], large effect; BF SRT $= 0.30 \pm 0.03$ s, SH SRT $= 0.27 \pm 0.03$ s) when compared to SH running. However, this effect was not statistically significant ($p > 0.05$). Figure 3 illustrates average SRT values for the familiarization trials and each experimental condition, with an increase in SRT indicating a decrement in SRT performance. When comparing the average SRT for the familiarization trial ($0.25 \pm 0.03$ s), SH running ($0.27 \pm 0.02$), BF running ($0.28 \pm 0.03$ s), and combined trials across both conditions ($0.28 \pm 0.02$ s), there was no statistically significant main effect of Condition ($F_{(3,21)} = 2.944$ $p = 0.057$).

Figure 4 demonstrates SRT errors during BF and SH running trials. There were no SRT errors during familiarization periods. SRT errors represented 17.6% and 17.0% of total SRT trials under the BF and SH running conditions, respectively ($17.6 \pm 6.6$ trials vs.

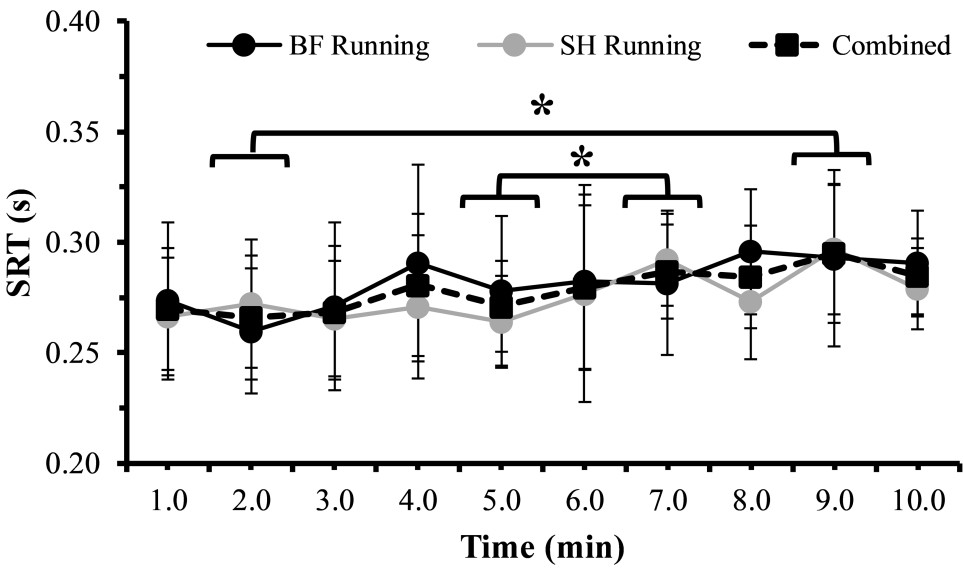

**Figure 2 Mean SRT from 1 min to 10 min during barefoot (BF Running, black circles with solid line) and shod (SH Running, gray circles with solid line) running conditions, as well as averaged across exercise conditions (Combined, black squares with dashed line).** Increasing SRT (s) indicates diminished SRT performance. A statistically significant main effect of Time ($F_{(9,63)} = 3.097$, $p = 0.004$) was found when assessing SRT. Statistically significant increases in SRT were present at 7 min relative to 5 min (0.29 ± 0.02 s vs. 0.27 ± 0.02 s, $p < 0.05$, $d = -0.99$, 95% CI [−1.98–+0.09], large effect), and at 9 min relative to 2 min (0.29 ± 0.03 s vs. 0.27 ± 0.03 s, $p < 0.05$, $d = -0.67$, 95% CI [−1.63–+0.37], moderate effect). Vertical bars represent one SD. Asterisks (∗) denote statistically significant differences ($p < 0.05$) at 7 min vs. 5 min, as well as at 9 min vs. 2 min.

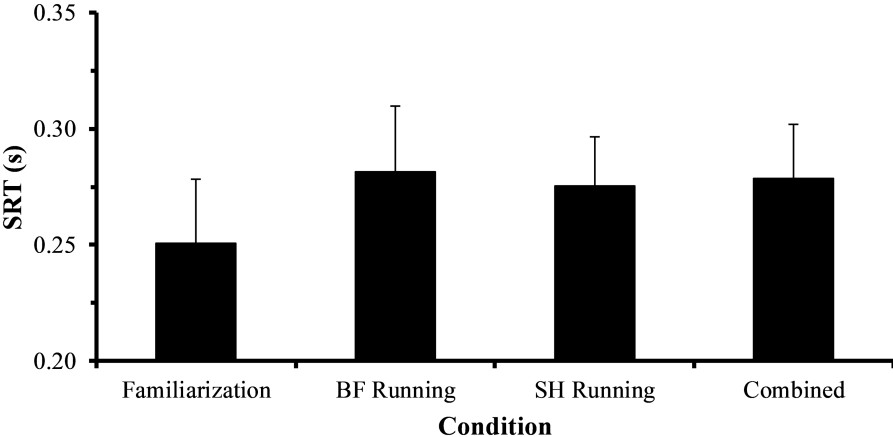

**Figure 3 Average SRT values for familiarization, barefoot running (BF Running), and shod running (SH Running) trials, as well as averaged across exercise conditions (Combined).** When the average SRT values across familiarization trials (0.25 ± 0.03 s), SH running (0.27 ± 0.02), BF running (0.28 ± 0.03 s), and combined exercise trials (0.28 ± 0.02 s) were compared using a one-way (Condition) rmANOVA, the main effect of Condition was not statistically significant ($F_{(3,21)} = 2.944$, $p = 0.057$). Increasing SRT (s) indicates diminished SRT performance. Vertical bars represent one SD.

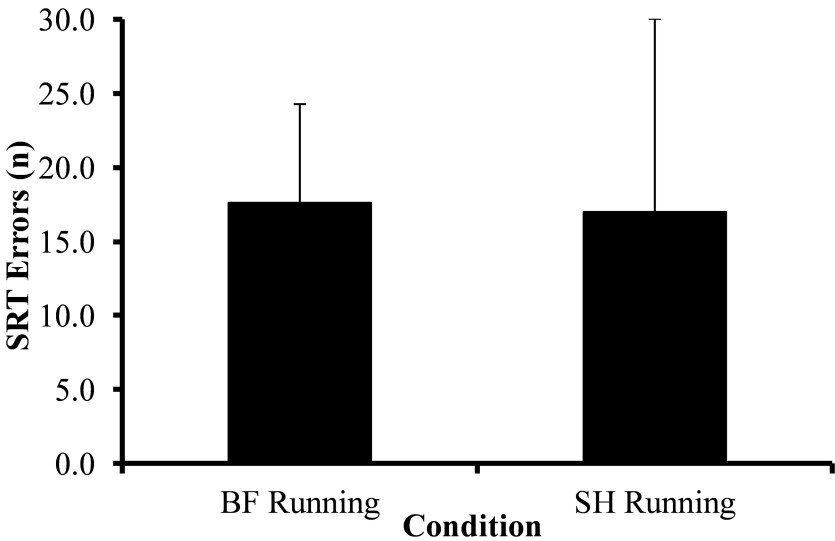

**Figure 4** **Average SRT errors across barefoot (BF) and shod (SH) running trials.** Absent SRT responses represented 17.6% and 17.0% of total SRT trials under the BF and SH running conditions, respectively (17.6 ± 6.6 trials vs. 17.0 ± 13.0 trials). The difference in SRT errors across conditions was not statistically significant ($t_{(7)} = 1.07$, $p = 0.918$). Vertical bars represent one SD.

17.0 ± 13.0 trials). There was no statistically significant difference in SRT errors across conditions ($t_{(7)} = 1.07$, $p = 0.918$).

Finally, seven participants reported feeling an increase in attentional demands during BF relative to SH running, while one participant noted no difference. In general, seven of eight participants (88%) highlighted: (i) a need to focus more on their footfalls to prevent uncomfortable landings; (ii) a perceived change in foot-strike patterns; and (iii) pain or burning on the plantar surface of the foot.

## Stride frequency

Stride frequency showed a statistically significant increase during BF running (88.3 ± 5.6 strides min$^{-1}$), relative to SH running (86.1 ± 5.7 strides min$^{-1}$, $p < 0.05$, $d = 0.39$, 95% CI [−0.62 to +1.36], small effect).

## DISCUSSION

The primary aim of the present study was to determine whether there was a difference in SRT during acute bouts of BF and SH running, in competitive distance runners naïve to BF running. Despite a statistically significant increase in SRT during later time-points of the exercise bouts compared to earlier, we did not observe a statistically significant difference in SRT across footwear conditions. We also did not find a statistically significant difference in SRT errors between conditions. However, there was a statistically significant increase in stride frequency during BF running; and participants anecdotally reported having perceived an increase in attentional demands during BF relative to SH running.

## SRT and attentional demands

Reaction time is an ecologically-relevant measure of perceptual-motor cost of running (*Schmidt & Lee, 2005*), due to the high prevalence of acute injuries sustained during running (*Hsu, 2012*; *Knobloch, Yoon & Vogt, 2008*), combined with the common nature of situations requiring reactions to extrinsic stimuli (*Magill, 2011*). In some instances, individuals must react quickly and suddenly to an unexpected stimulus to avoid injury, making it imperative to avoid any threat to reaction time performance (*Magill, 2011*; *Schmidt & Lee, 2005*). On this basis, increased SRT has been linked to a possible increased injury risk during running, due to falls and sprains (*Beynnon et al., 2001*; *Konradsen & Ravn, 1990*; *Konradsen & Ravn, 1991*; *Murphy & Connolly, 2003*; *Richardson et al., 2017*; *Willems et al., 2005*). Past work highlights that runners need to pay more attention to their foot-strikes during BF running (*Alloway et al., 2016*), and to alter their running kinematics to avoid noxious plantar stimuli (*Lieberman et al., 2010*). Consequently, we hypothesized that BF running would produce a detrimental effect on SRT performance, with reference to SH running; yet we observed no statistically significant effect of BF running on SRT or SRT errors.

In the present study, participants anecdotally reported an increase in attentional demands during BF relative to SH running, noting: (i) a need to focus more on their footfalls to prevent uncomfortable landings; (ii) a perceived change in foot-strike patterns, particularly during the latter minutes of BF running; and (iii) pain or burning on the plantar surface of the foot. A recent study directly comparing BF and SH running trials on an indoor track showed that runners had to pay greater attention to their foot-strikes during the BF condition, as evidenced by greater working memory when stepping on targets during running (*Alloway et al., 2016*). This observation is supported by work that has indicated that SRT performance is decreased in the presence of externally applied cutaneous stimulation (*Hanson, Whitaker & Heron, 2009*). When considering the concept of dual-task interference, which emphasizes participants' limited attentional capacity (*Watanabe & Funahashi, 2017*), it could be expected that cognitive task performance would suffer in the presence of an attentionally demanding procedure such as BF running. Therefore, at the same relative intensity, it is possible BF running does not have any additional attentional demand compared to SH running. It is also plausible that by providing participants with rubber-gripped toe socks for the BF running condition, plantar sensory feedback was masked relative to a true BF running condition, introducing a potential confound to our experiment. However, given runners' subjective comments, we consider this possibility unlikely.

## Stride frequency

The statistically significant increase in stride frequency with BF running may reflect kinematic differences between BF and SH running (*Divert et al., 2005*; *Ekizos, Santuz & Arampatzis, 2017*). Increased stride frequency could be the result of increased somatosensory feedback present during BF running and intended to avoid painful foot-strikes (*Hsu, 2012*; *Lieberman, 2012*). Indeed, increased sensory feedback during BF running can alter foot-strike patterns, for instance by modulating ankle coordination

prior to foot-strike (*Kurz & Stergiou, 2004*), reducing ground reaction forces (*Lieberman et al., 2010*), and increasing and decreasing stride frequency and contact time, respectively (*Ekizos, Santuz & Arampatzis, 2017*; *Kurz & Stergiou, 2004*). During SH running, this protective feedback is believed to be impaired (*Robbins, Waked & McClaran, 1995*). We anticipated that such differences in somatosensory feedback would have a limiting effect on attentional capacity, reflected by an increase in SRT. Yet, despite the statistically significant increase in stride frequency during BF running, SRT did not change.

## Possible explanations

Despite anecdotal reports and past research suggesting a potential attentional difference between BF and SH footwear conditions, we did not observe such an effect. There is evidence in support of our finding, that SRT was not different during BF versus SH running. For example, *Klint et al. (2008)* showed that proprioceptive and cutaneous feedback from the foot and leg can modulate stepping patterns and increase variation in kinematics during BF locomotion, independent of higher processing. They concluded that higher centers are likely reserved for more complex movements. Others have intimated that gait involves the integration of spinal activity, afferent sensory information, and efferent motor commands from the primary motor cortex and pyramidal tract, with reactions to external perturbations being integrated at the cortical or at the spinal level, depending on the nature of the perturbation (*Nielsen, 2003*). *Nielsen (2003)* proposes that concomitant spinal and supraspinal regulation of gait allows for more flexible integration of sensory (e.g., visual, somatosensory) and motor information online. This is likely why neural interactions governing gait modulation are rapid, allowing stride-to-stride variation in stepping patterns (*Klint et al., 2008*; *Nigg & Wakeling, 2001*).

Greater variation of stride kinematics during BF running might therefore be associated with injury prevention and pain reduction, due to a reduction in repeated impact (*Lieberman et al., 2010*). However, if the increased afferent feedback experienced during BF running does not undergo higher processing (*Klint et al., 2008*), then runners would be at no greater risk of acute injuries due to stumbles or falls (*Beynnon et al., 2001*; *Konradsen & Ravn, 1990*; *Konradsen & Ravn, 1991*; *Murphy & Connolly, 2003*; *Richardson et al., 2017*; *Willems et al., 2005*). This is supported by our finding that BF running did not have a statistically significant effect on SRT. Although speculative in nature, this lack of conscious processing of somatosensory and proprioceptive information would serve a protective role for persons engaging in BF running.

## Methodological considerations and future directions

There are a few noteworthy methodological considerations in the current work which could have influenced our findings. Throughout the course of prolonged exercise fatigue can negatively influence corticospinal and neuromuscular output (*Meardon, Hamill & Derrick, 2011*; *Ross et al., 2007*), and consequently reduce perceptual-motor performance (*Brisswalter, Collardeau & René, 2002*). The present results support existing evidence in that SRT performance tended to increase towards the end of the exercise bout, a result that has been previously reported in the literature (*Brisswalter et al., 1995*;

*Brisswalter, Collardeau & René, 2002*; *Collardeau, Brisswalter & Audiffren, 2001*; *McMorris & Keen, 1994*). However, this was a short and moderately-intense bout of steady-state exercise (*American College of Sports Medicine, 2013*), with an appropriately-timed rest period (*Saltin et al., 1968*); so, fatigue was not likely a major contributor to decreased performance (*Lambourne & Tomporowski, 2010*; *Tomporowski, 2003*). Indeed, the exercise duration and intensity were chosen to coincide with a level of physiological arousal that optimizes SRT (*Brisswalter, Collardeau & René, 2002*; *Collardeau, Brisswalter & Audiffren, 2001*; *Lambourne & Tomporowski, 2010*; *Tomporowski, 2003*). However, given we did not measure HR or perceived exertion throughout SRT testing, it is possible that participants' level of exertion during exercise was greater than the exercise intensity prescribed. Further work should examine HR and perceived exertion during exercise and SRT testing. Aside from exercise intensity, the decrease in performance observed may have been a result of the mode of exercise (i.e., treadmill exercise), as compared to exercise intensity or fatigue (*Lambourne & Tomporowski, 2010*). In other words, given the dual-task nature of the present experiment, SRT performance may simply have suffered in response to participants' avoiding falling off the treadmill. Nevertheless, there was no statistically significant difference in SRT during resting familiarization trials compared to exercise. In addition, without the use of treadmill running, it would not have been possible for us to employ the present SRT task.

Secondly, our measure of SRT was unable to decompose participants' overall response time, into its constituent components of reaction time (i.e., time between stimulus presentation and initiation of response), which measures stimulus detection; and movement time (i.e., time between response initiation and response completion), which measures response execution (*Magill, 2011*). Past work has shown that acute exercise preferentially influences movement time over reaction time in SRT tasks (*Beyer et al., 2017*; *Davranche et al., 2005*; *Davranche et al., 2006*), indicating that exercise-induced changes in response time are related more to faster movement execution than changes in cognitive function (*Beyer et al., 2017*). Thus, to elucidate the cognitive influence of BF running it would be prudent for further work to examine a greater number of dimensions of task performance, including separating reaction and movement times. Similarly, examining more complex cognitive tasks (e.g., discrimination RT) may better discern the cognitive impacts of BF running, as opposed to SRT which simply examines speed of information processing (*Alloway et al., 2016*; *Beyer et al., 2017*). Finally, it is conceivable that the small sample size in this exploratory study may have threatened the validity of the observed results. Likewise, by including a sample of male runners exclusively, our findings may not be generalizable to female athletes. Consequently, future work will benefit from examining a larger sample of runners, including both females and males, and recreational runners who have no experience using minimalist footwear. Subsequent investigations should also sample a larger number of SRT trials over a longer time-period, with measurements of physiological and perceived exertion.

## CONCLUSIONS

In the present exploratory study, an acute bout of BF versus SH running did not impact SRT. It is possible that increased afferent feedback during BF running (*Kurz & Stergiou, 2004*; *Robbins et al., 1993*) is responded to in subcortical regions or transcortical reflex pathways (*Nielsen, 2003*), without affecting the attentional requirements of the task. Additionally, this may be the case only for simple tasks such as SRT. Alternatively, it is possible that our small sample size did not have sufficient power to reveal a significant difference across BF and SH running conditions. Nevertheless, the present results suggest that although differences in running kinematics across BF and SH running may lead to differences in musculoskeletal injuries (*Daoud et al., 2012*; *Hreljac, 2005*), it is not likely that BF running will impact runners' risk of attention-related acute injuries such as trips or falls (*Hsu, 2012*; *Knobloch, Yoon & Vogt, 2008*). Future work should examine whether more complex perceptual-motor tasks and more sensitive outcomes will be affected by BF versus SH running. Further efforts should also examine whether the present observations will emerge in a larger sample of runners. Finally, it is prudent to examine whether changes in SRT will manifest when runners are performing over-ground on a stable running surface, as opposed to during treadmill running.

## ACKNOWLEDGEMENTS

The current study received no external funding support, and the authors have no conflicts of interest to declare. We wish to sincerely thank Mr. Blaise Dubois for his tremendous support during experimental planning, Dr. Normand Teasdale for his generous assistance in reviewing the original manuscript before its initial submission, and Dr. Thamir Alkanani for his technical contributions.

### Funding

The authors received no funding for this work.

### Competing Interests

The authors declare there are no competing interests.

### Author Contributions

- Nicholas J. Snow, Jeannette M. Byrne and Fabien A. Basset conceived and designed the experiments, performed the experiments, analyzed the data, contributed reagents/materials/analysis tools, prepared figures and/or tables, authored or reviewed drafts of the paper, approved the final draft.
- Jason F.L. Blair performed the experiments, analyzed the data, contributed reagents/materials/analysis tools, authored or reviewed drafts of the paper, approved the final draft.

- Graham Z. MacDonald conceived and designed the experiments, analyzed the data, contributed reagents/materials/analysis tools, authored or reviewed drafts of the paper, approved the final draft.

## Human Ethics

The following information was supplied relating to ethical approvals (i.e., approving body and any reference numbers):

The Interdisciplinary Committee on Ethics in Human Research (ICEHR) from Memorial University granted Ethical approval to carry out the study within its facilities (Ethical Application Ref: 20130246-HK).

## Data Availability

The raw data are provided in Data S1.

## Supplemental Information

Supplemental information for this article can be found online at http://dx.doi.org/10.7717/peerj.4605#supplemental-information.

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
