# Peer review of "Barefoot running does not affect simple reaction time: an exploratory study"

_PeerJ, doi:10.7717/peerj.4605_

## Round 0.1 · original submission · Minor Revisions

The manuscript is well written, good scientific quality/rigor with an appropriate number of references to support it. The Background is will written and forms the basis for the research. Statistical analyses are thorough and appropriate.

There are a small number of edits required, typographical error and references that must be corrected thru your citation/reference software. Additionally, your figures are of low quality (blurry) and will need to be improved.

Additional specific comments are included on the attached documents.

I congratulate the authors on the research completed.

·

Basic reporting

1. Basic Reporting
A. Clear and unambiguous, professional English used throughout.
Meets criteria.
B. Literature references, sufficient field background/context provided.
Introduction provides sufficient connectivity to the current literature with adequate references.
C. Professional article structure, figs, tables. Raw data shared.
Resolution of Figures 1, 2, and 3 are fuzzy (noting this maybe an artifact of converting to a PDF file). Resolution of figure 4 is acceptable. Data is available and readable.
D. Self-contained with relevant results to hypotheses.
Yes.

Experimental design

2. Experimental design
A. Original primary research within Aims and Scope of the journal.
Yes.
B. Research question well defined, relevant & meaningful. It is stated how research fills an identified knowledge gap.
Yes and does so in a compelling fashion.
C. Rigorous investigation performed to a high technical & ethical standard.
Study conducted with permission from Interdisciplinary Committee on Ethics in Human Research at the Memorial University of Newfoundland (ethics approval number: 20130246-HK).
D. Methods described with sufficient detail & information to replicate.
The authors did an outstanding job in this regard.

Validity of the findings

3. Validity of the findings
A. Impact and novelty not assessed. Negative/inconclusive results accepted. Meaningful replication encouraged where rationale & benefit to literature is clearly stated.
Novelty was accessed, negative results were reported and related to existing literature. Replication was discussed within meaningful parameters.
B. Data is robust, statistically sound, & controlled.
Statistical and data analysis are appropriate to answer the research question and supported the research design employed.
C. Conclusion are well stated, linked to original research question & limited to supporting results.
Yes.
D. Speculation is welcome, but should be identified as such.
I did not detect any unwarranted speculation.

Additional comments

4. General comments
I have been reviewing papers for over 15 year now. I must say, this is in the top 95% of well written papers I have come across! Cheers to the authors!
With that said, there are some minor comments embedded in the manuscript via the tracker function. Most of the comments are typos and such. There are a couple of content comments that should be addressed and/or included prior to publication.
5. Confidential notes to the editor
See tracker comments embedded in manuscript.

Reviewer 2 ·

Basic reporting

This study was very well written and thorough. Literature review was complete and provided strong basis for the study and sound explanation of results.

Figures 1, 2, and 3 seemed a bit blurred but content was relevant.

Experimental design

The research question was clearly articulated. The research design was especially strong, especially in light of controlling for confounding variables. The methods were very well detailed. The only potential downsides were the small subject number but as an exploratory study, this was explained. Also, only males were used and no explanation was provided as to why the subject population was limited to one gender (Line 125). Additional justification would be appropriate here.

Validity of the findings

The results were well presented and the subsequent discussion was very sound. The discussion was well referenced and clear. This was a strong suit of the paper.

Additional comments

This was a very well designed and well written study. It would be good to expand and include females, if possible but overall this was a strong paper.

---

## Round 0.2 · accepted · Accept

On behalf on the Reviewers I would like to congratulate the authors for addressing all of the concerns and edits raised in a very timely manner. Congratulations again on your manuscript and thank you for your contribution to PeerJ.